

**Short Communication: A database of the global distribution of (U-Th)/He ages and U, Th contents of goethites**

Hevelyn S. Monteiro[1*], Kenneth A. Farley[1], Paulo M. Vasconcelos[2]

1. California Institute of Technology, Pasadena, CA, 91125, USA

2. The University of Queensland, Brisbane, QLD,4072, Australia

*Correspondence to: hevevelynbr@gmail.com*

**Abstract**. Terrestrial supergene goethites of known ages record information on changes in weathering conditions through
time. Here we present a database of (U-Th)/He ages and U and Th contents of goethites from different weathering environments around the globe. By consolidating published data collected at four different laboratories and unpublished data collected at the Noble Gas Laboratory at Caltech, we aim to give an overview of the work carried out by geochronologists and geochemists in the last 20 years. The database contains 2597 (U-Th)/He ages of goethites from 10 countries; most of the ages come from Brazil and Australia.

## 1. Introduction

Goethite (α-FeOOH) records information on water-rock interactions in near surface environments (e.g., Yapp, 2001). Supergene goethites potentially reveal how old weathering profiles are, how meteoric solutions evolved during the weathering process and through time, and how fast or slow chemical denudation transforms rocks in the subsurface.
Attempts to date supergene goethite date back to Strutt (1910). More recently, Lippolt et al. (1998) revisited the (U-Th)/He method to determine the ages and evaluate He retentivity in supergene goethites. Shuster et al. (2005) used the $^4$He/$^3$He method to quantify He retentivity in various types of goethites, showing that (U-Th)/He results could be corrected for He losses and that well-crystallized goethite retained more than 90% of its He for millions of years. The combined (U-Th)/He-$^4$He/$^3$He method was used to date goethites from lateritic profiles in Brazil and one sample from Mount Isa, Australia
(Shuster et al., 2005). Since 2005, numerous studies have confirmed that (U-Th)/He geochronology of goethite from various settings sheds light on changes in global environmental conditions.

The application of (U-Th)/He dating to goethite-bearing weathering profiles across the globe has provided new insights into the Earth's surficial history. Studies in Western Australia (e.g., Heim et al., 2006; Vasconcelos et al., 2013; Danisik et al., 2013; Yapp and Shuster, 2017) and Brazil (e.g., Lima, 2008; Monteiro et al., 2014, 2018(a,b), 2022; Conceição et al., 2024)
have unveiled a protracted weathering history, showing that goethite age distribution depends on climate and erosional





history. Similarly, studies in French Guiana (Heller et al., 2022), Suriname (Ansart et al., 2022), Morocco (Verhaert et al., 2022), and Tunisia (Yans et al., 2021) further show the influence of paleoclimate on weathering and supergene ore genesis. Dated pisoliths from the Bohnerz Fomation paleosol in Central Europe (Hofmann et al., 2017) show intensification of weathering and soil formation during the Miocene. These studies, and many others compiled here, highlight the utility and
applicability of goethite (U-Th)/He geochronology.

Here we compile a global database of goethite distribution, (U-Th)/He ages, and U and Th concentrations, the latter a byproduct of the age determination. From this global compilation we infer the main factors controlling the formation and preservation of supergene goethite. Building upon the findings from each individual study, we aim to assess the influence of environmental conditions on changes in the frequency of precipitation and preservation of goethites in weathering
profiles.

## 2.   Goethite U-Th-He database

The database (https://github.com/hevelyn-monteiro/GlobalGoethite_U-Th-He_Ages.git) comprises 2597 goethite (U-Th)/He ages, with 2362 U (ppm) and 2358 Th (ppm) measurements. Most of these ages – 2161 – come from published
studies, while the remaining are our unpublished measurements made at Caltech over the last two decades using methods described in Shuster et al. (2005) and Monteiro et al. (2014, 2018a). Some authors report U and Th measurements in units such as ng, nmol, or nmol/g. For cases where the mass of the analyzed grain was not provided and parent element amounts are reported, it was impossible to calculate concentrations in ppm. Consequently, only the ages of these samples are summarized here. In addition to U and Th concentrations and (U-Th)/He ages, the database also contains information on
geographical location, elevation (m), sample depth (m), bedrock, Sm (ppm) and eU (ppm) concentrations, and Th/U values. Effective uranium (eU) is defined as $(U + 0.235 Th)$ and is a single metric that approximates the alpha particle production rate.

Note that many entries in the database represent analyses of goethite subsamples from a single hand sample. In some cases, these may represent replicates of a single generation of goethite, while in others they may represent distinct generations. In
the discussion below we make no attempt to weight results for possible duplicate analyses; every analysis is considered an independent result.

Figure 1 illustrates the map distribution of (U-Th)/He dated goethites. Most samples are from Brazil [1428] and Australia [472]. This dominance reflects the interests of the researchers driving the studies, the abundance of supergene goethite in laterites and ferricretes in the two countries, and the ease of access to sampling sites in open-pit mining operations. The two
countries are major iron ore producers, and it makes sense to search for iron minerals in these locations. In addition, Brazil and Australia sit at similar geographic positions near the tropics but contrast greatly in present and paleo climates and



Cenozoic plate motion paths, providing excellent test grounds for assessing the relative roles of climate and tectonics on the formation and preservation of goethites in the landscape. (U-Th)/He ages for goethites are also available from Switzerland [195], Suriname [193], French Guiana [134], Morocco [55], China [52], USA [41], Tunisia [14], and Canada [6]. Goethites

from different environments provide information on geological processes of local and global significance. Therefore, we will summarize the most important characteristics of each type of geological environment from which ages of goethites have been obtained so far, along with the different types of goethites investigated by (U-Th)/He geochronology.

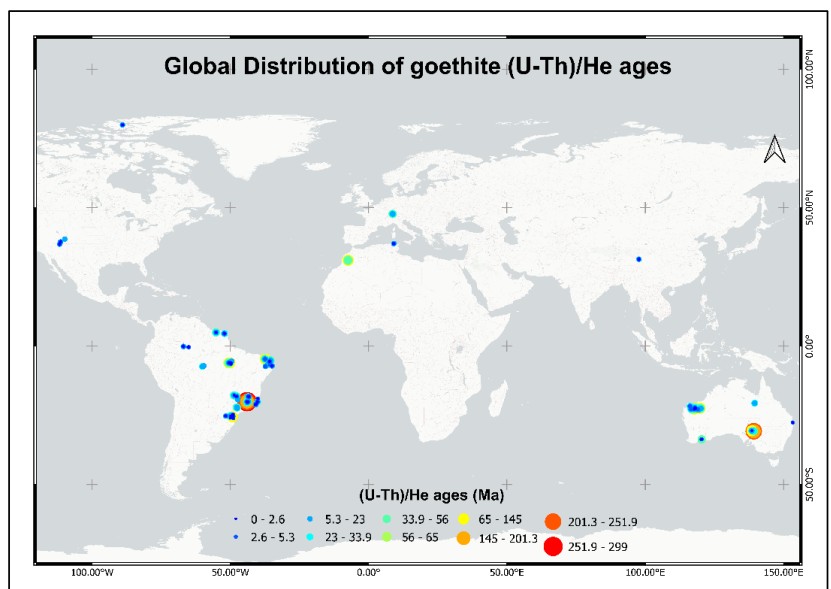

**Figure 1 illustrates the geographical distribution of dated goethites worldwide. A total of 2597 (U-Th)/He ages were compiled from published and unpublished research. The large majority of dated goethites come from Brazil (55%) and Australia (18%). Globally, the largest number of goethites are of Miocene age (40%), while the second (20%) and third (16%) largest groups of dated goethites fall within the Pleistocene and Pliocene epochs, respectively. (U-Th)/He ages older than 65 Ma only occur in Brazil (Amazon and Quadrilátero Ferrífero), Australia (Hamersley Province and Flinders Ranges), and Morocco.**


### 3.  Geological Environments

Goethites form in oxidizing, acidic to alkaline environments within a narrow – 0 to ~70 °C – range of temperatures at the Earth's surface and shallow subsurface (e.g., Yapp, 2001). Their composition and crystallinity are determined by the geological environment and mode of precipitation. In the lithosphere-hydrosphere-atmosphere-biosphere interface occupied

by weathering profiles, the main process linked to iron duricrust formation is goethite precipitation-dissolution-





reprecipitation, which favors the preservation of primarily young (< 3 Ma) goethites even in old landscapes (as old as 70 Ma) (Monteiro et al., 2014, 2018a). Goethites precipitated deep within a weathering profile interact less frequently with organic acid-rich weathering solutions that drive iron dissolution-reprecipitation in the near surface environment and often preserve older results (Monteiro et al., 2018a, b). The abundance, chemical composition, morphology, and age distribution

of goethite reflect its precipitation environment. Below is a summary of the most important characteristics of each geological environment hosting weathering profiles containing goethites included in our geochronology compilation.

### 3.1.  Lateritized banded iron formation

Banded iron formations (BIF) are rich in Fe, but poor in U and Th (Spier et al., 2007; Pecoist et al., 2009). Weathered banded

iron formations in Brazil and Australia are commonly capped by goethite-cemented duricrusts (cangas), providing an abundance of goethite for geochronology (Shuster et al., 2012; Monteiro et al., 2014, 2018b). Early studies of goethite from BIFs aimed to evaluate the suitability of well-crystallized and relatively U-Th-poor goethite for (U-Th)/He geochronology. Hematite and magnetite, and in some cases iron-bearing carbonates, are the primary sources of iron released during BIF weathering. As other co-existing minerals (mostly quartz, chalcedony, and calcite, with minor apatite, talc) are dissolved

and their elements are leached away, iron contents increase progressively (sometimes exceeding 65% Fe). Soluble $Fe^{2+}$ released from slowly reacting magnetite and reductive dissolution of hematite and goethite locally migrates in solution, oxidizes, and reprecipitates as goethite.

Over tens of millions of years, weathering reactions led to the formation of deep lateritized BIFs globally, with notable examples in the Quadrilátero Ferrífero and Serra dos Carajás regions, Brazil and the Hamersley Province, Australia. The

stratigraphy of these lateritic profiles comprises a goethite-cemented duricrust (canga), an *absolutely* iron enriched weathered BIF layer underlain by a *relatively* enriched zone that transition into a saprock and eventually into the unweathered BIF at depth (Samama, 1986). Goethite dominates the mineralogy of cangas (Monteiro et al., 2014). In contrast, primary hematite inherited from the bedrock dominates in the lower horizons, where goethite is restricted to veins or local replacement of minor carbonates or silicates (e.g., siderite or grunerite) and they generally record old weathering

ages (Monteiro et al., 2018a). Near the surface, the canga forms an indurated brecciated cap hosting fragments of BIFs and hypogene hematite-magnetite ore cemented by multiple goethite generations (Monteiro et al., 2014). Canga resides close to the surface, where abundant $O_2$ and air-rich meteoric solutions result in a dominantly oxidizing environment; however, vegetation and microorganisms release enough organic acids to enhance the dissolution of $Fe^{3+}$-bearing minerals, leading to recurrent events of iron dissolution-reprecipitation (Monteiro et al., 2014; Levett et al., 2020). Iron dissolution-

reprecipitation protects the duricrust against physical erosion because broken fragments are quickly recemented and stabilized – the self-healing property defined by Monteiro et al. (2014). In addition to immobilizing iron, canga goethites incorporate many elements introduced laterally from nearby rocks and from above (e.g., dust) and are enriched in Al and



Th (Monteiro et al., 2018a). In contrast, U is leached downward into the saprolite producing U-rich goethites in the absolute or relatively enriched environments (Monteiro et al., 2018a). Recurrent iron dissolution-reprecipitation results in goethites displaying large age spans but dominance of younger generations ($\leq \sim 3$ Ma).

### 3.2. Channel iron deposits

Channel iron deposits (CID) are a unique iron mineral occurrence of great economic significance (Ramanaidou et al., 2003 and references therein). During the Cenozoic, broad meandering rivers carved the BIF landscape in Western Australia. Massive erosion events in the late Eocene resulted in the production and delivery of significant quantities of sediments into these meandering rivers, aggrading the channels (Vasconcelos et al., 2013). Aggraded sediments included detrital BIF fragments, previously formed ferricrete detritus, some silicates and clay minerals locally eroded from shales and dykes, and wood fragments (Morris et al., 1993). The mixture of iron-bearing minerals, interstitial water, and organic matter created the conditions for widespread iron dissolution-reprecipitation, cementing and Fe-metasomatizing the sediments producing massive beds of pisolitic goethites, now iron ore. Three CIDs from the Hamersley Mineral Province, WA, have been dated by the (U-Th)/He method: Yandicoogina CID (Heim et al., 2006), Lynn Peak CID (Vasconcelos et al., 2013), and Robe River CID (Danisik et al, 2013; Yapp and Shuster, 2017). (U-Th)/He ages show that iron cementation of the sediments started ~33 Ma and intensified during the Miocene; notably, CID goethites younger than ~5 Ma have not been documented.

### 3.3. Gossans and lateritized Fe-Cu-Au deposits, Carajás Mineral Province, Brazil

Weathered massive sulfide deposits are another example of supergene systems that offer a wealth of goethite varieties. The Carajás region, Brazil, hosts numerous iron-oxide copper gold (IOCG) deposits (e.g., Igarapé Bahia, Salobo, Sossego) that are variably weathered. Deposits hosted within the 700-800 m Carajás plateau are weathered to >100 m depths; deposits hosted within the surrounding ~300 m-elevation Itacaiunas Surface are weathered to 20-40 m (Monteiro et al., 2018b). Complete weathering profiles are exposed by open pit mining operations in both landscape positions, providing access to abundant samples. Lateritized massive sulfide deposits hosted by the Carajás plateaus are covered by clay-rich soils (1-15 m) underlain by thick (up to 20-30 m) Fe-Al duricrust, mottled zone, saprolite, and gradational or sharp transitions into bedrock at depths ranging from 90 to > 120 m (Monteiro et al., 2018b). Gossans are often present, particularly in the weathered zones directly overlying massive sulfide concentrations at depth. During evolution of the weathering profiles, $Fe^{2+}$ and $Cu^{2+}$-rich acid solutions produced by sulfide weathering moved downward until the solutions were neutralized by reaction with wall-rock silicates and carbonates, driving precipitation of Cu ions as secondary phases – e.g., malachite, azurite, cuprite, native copper, and chalcocite – and Fe as goethite. While copper was more effectively leached and enriched



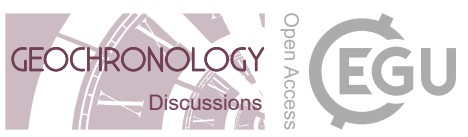

towards the bottom of the profile, goethites formed throughout the entire system. In the soil layer, goethite is abundant (~30

wt%) but too fine grained and not suitable for geochronology. Goethites from the duricrust are commonly enriched in Al
and often occur in massive blocks composed of several cross-cutting generations. They may also grow as colloform masses
around root casts or insect burrows (Monteiro et al., 2018b). Within and immediately below the duricrust, goethite often
forms miniature stalactites and stalagmites crowned with euhedral gibbsite and gold crystals. Within the saprolite and
saprock, pure colloform goethites suitable for geochronology may occur in veins or vugs or larger cavities produced by

sulfide weathering. Goethites precipitated at depth tend to incorporate percentage amounts of Cu, Mn, and P in solid
solution; they are also very rich in U, but Th depleted. Colloform goethites tens of centimeters in size are common. Ages of
goethites (Shuster et al., 2005; Monteiro et al., 2018b) from gossan and saprolite in the Igarapé Bahia Cu-Au deposit in
Brazil confirm the prolonged history (70-1 Ma) of weathering of the Carajás plateau obtained from nearby lateritic iron and
Mn deposits (Shuster et al., 2012; Vasconcelos, 1999a).


### 3.4.    *Nickel laterite, Ravensthorpe, Western Australia*

Weathering of dunites, harzburgites, serpentinites, komatiites, some layered mafic intrusions, or nickel sulfide deposits
produce nickel laterites (Golightly, 1981). The descending oxidizing weathering solutions promote rapid breakdown of

mafic minerals in the protores, and weathering solutions become progressively less acidic with depth (Golightly, 1981).
Depending on the parental rock, nickel laterites may comprise an iron or leached siliceous duricrust, a limonite horizon, a
clay-rich (smectite) horizon, saprolite, saprock, and bedrock (Samama, 1986). The Ravensthorpe nickel laterite in Western
Australia originates from mafic-ultramafic sequences of the Ravensthorpe greenstone belt (Mostert, 2014). The laterite has
an average thickness of 60 m. Goethites showing various degrees of crystallization occur in all horizons of the Ravensthorpe

Ni-laterite; they probably formed under neutral to alkaline conditions resulting from weathering of ultramafic rocks. (U-
Th)/He ages of goethites from the Ravensthorpe Ni-laterite range from ~5 to ~1 Ma.

### 3.5.    *Lateritized alkaline-carbonatite complexes*

Alkaline-carbonatite complexes (ACC) host a variety of rock types (e.g., glimmerites, phoscorites, carbonatites) and diverse
mineralization (Nb, P, REE, Ti, Cu, Ni, etc.) (e.g., Simandl and Paradis and reference therein). The average concentrations
of U and Th in alkaline magmas are 10 and 35 ppm, respectively (Wedepohl, 1978). However, apatite, monazite, and
xenotime in alkaline-carbonatite rocks contain hundreds of ppm U and Th, in addition to economic concentrations of other
REEs. In the Alto Paranaiba Igneous Province, Brazil, lateritized alkaline-carbonatite complexes (e.g., Catalão, Araxá,





Tapira, Poços de Caldas) form ~100-meter-thick saprolites exposed at elevated dome structures protected from erosion by the fenitization of surrounding quartzites (Conceição et al., 2022). Goethites resulting from weathering of alkaline-carbonatite rocks provide information on the evolution of supergene mineralization in these systems, and their unique geological environments, protected from erosion, record protracted histories of weathering. Ages of goethites from the Alto Paranaiba Igneous Province vary from ~40 to < 1 Ma (Marques et al., 2023; Conceição et al., 2024).


### 3.6. Lateritized basalts and acid to intermediate igneous rocks

When lateritized, basalts and acid to intermediate igneous rocks produce chemically stratified weathering profiles composed of soils, ferricretes, bauxites, mottled zones, saprolites, saprock, and bedrock (Samama, 1986). The complex texture observed in ferricretes and bauxites formed on extrusive igneous precursors arises from the co-existence of different types

of oxyhydroxides – e.g., goethite, hematite, maghemite, ilmenite, magnetite, rutile, gibbsite, boehmite – and clay minerals in crusts, pisoliths, cements, and from the iron metasomatism and pseudomorphic replacement of silicates. Al- and Ti-rich goethites are common in these weathering profiles. Refining the geochronology of goethites and hematites from weathered basalts on Earth provides guidance for targeting samples at Mars that may be suitable for investigating water-rock interaction

in the early history of that planet. In southern Brazil, goethites from weathered basalts underlying the Third Paraná Plateau are younger than 6.2 Ma (Riffel et al., 2016). Heller et al. (2022) present goethite (and hematite) (U-Th)/He ages for duricrusts blanketing lateritized metavolcanic rocks in French Guiana and show that weathering started in the Oligocene and it intensified after ~6 Ma. Further investigation of weathered basalts on a global scale is needed.

### 3.7. Lateritized continental sediments

Deeply weathered continental sediments, often of unknown age, commonly contain both detrital and authigenic iron oxides and hydroxides suitable for (U-Th)/He geochronology (e.g., Lima, 2008). Dating both detrital and authigenic goethites from these sediments permits bracketing the age of the sedimentary units, and it provides information on the weathering profiles

that were eroded to produce the sediments, and the post-depositional climatic conditions that promoted the in situ ferruginisation of the sedimentary units. For instance, the Barreiras Formation along the coast of Brazil (Mabesoone et al., 1972; Lima, 2008; Rossetti et al., 2011; Monteiro et al., 2022) forms coastal plateaus from the mouth of the Amazon River to Rio de Janeiro. These sediments host fragments of duricrusts, detrital and authigenic pisoliths, and in situ ferricretes (Monteiro et al., 2022). Post-deposition wet and warm conditions promoted widespread lateritization of the sediments

(Monteiro et al., 2022). The mobilization and accumulation of iron can be associated with variations of pH-Eh boundary



conditions by vertical fluctuation of the water table and the invasion of seawater, the release of organic acid by plants, and microbial activities (Monteiro et al., 2022). Lima (2008) dated coexisting detrital and authigenic pisolitic goethites from the Barreiras Formation, northeastern Brazil, to bracket the time of deposition and post-deposition weathering of the sediments. Monteiro et al. (2022) dated goethites from the Barreiras Formation in southeastern Brazil and determined that weathering conditions in NE and SE Brazil were initially similar but started to diverge after ~10 Ma (Monteiro et al., 2022).

In central and southwestern Amazon, (U-Th)/He dating of goethite from lateritized sedimentary units reveal an increase in goethite precipitation or preservation in the Pleistocene (Allard et al., 2013) and Early-Middle Miocene (Albuquerque et al., 2020). In central Europe, clay-rich soils hosting goethite pisoliths formed between ~50 and ~ 17 Ma (Hoffmann et al., 2017).

### 3.8. Karst environments

Karst environments are formed by the congruent dissolution of rocks – dolostones, quartzites – interacting with surface waters over thousands to tens of millions of years. Geological and environmental parameters such as lithology, permeability, degree of fracturing, relief, hydraulic gradient between recharge and discharge areas, and climate control karst development (Samama, 1986). (U-Th)/He ages of goethites precipitated in cavities in dolostones from the Imini Mn ore deposit, Morroco (Verhaert et al., 2021), reveal that iron-rich meteoric solutions reacted with carbonates during karstification in the Upper Cretaceous (95– 80 Ma). The goethites formed in the subsurface under predominantly alkaline conditions and relatively arid climates. These environmental conditions favored the preservation of these old goethites. Moroccan goethites, as a group, represent the oldest continuously exposed and in situ goethite population dated on Earth so far.

Supergene goethites are also present in quartzite karst across the planet (Wray et al., 2017). Supergene goethite from the Espinhaço Range, Minas Gerais, Brazil provides insights into the history of weathering and re-weathering possible when relatively weatherable lithologies (phyllites) occur interbedded with and scaffolded by chemically resilient quartzites (de Campos et al., 2023). In these systems, geochronology of goethites reveals a wealth of information about the paleoclimatic histories of continental interiors. (U-Th)/He ages obtained for three main groups of ochre, brown, and black vitreous goethites range from 18.3 to 7.8 Ma, 6.4 to 2.5 Ma, and 1.8 to < 1 Ma, respectively (de Campos et al., 2023), showing that the focused hydrology of the quartzite karsts drives recurrent precipitation-dissolution-reprecipitation of goethites in these systems.

### 3.9. Coal deposits



Goethite replacing a tree trunk was collected in a coal mine from the Springsure region, Queensland, Australia. Total preservation of the tree cells shows minimum compression of the fossilized trunk, suggesting that ferruginization preceded burial and coalification of associated vegetation. (U-Th)-He ages obtained for goethites collected parallel to the tree's growth axis range from ~120-90 Ma. These ages probably represent cooling ages associated with exhumation of the coal deposits.


## 4. Types of goethite

Goethites formed in different geological environments will vary in major, minor, and trace element contents; macroscopic characteristics (habit, crystallinity, color, porosity, luster); purity; stratigraphic position within the weathering profile; and probability of preservation. Goethite may form by direct precipitation of iron from solution, filling empty cavities and
fracture planes and often forming large colloform or botryoidal masses; via pseudomorphic replacement of primary Fe-bearing phases (e.g., pyrite, magnetite) forming massive goethite; iron metasomatism of sediments and sedimentary, igneous, or metamorphic rocks; precipitation in concentric layers forming pisoliths; through microbially-induced nucleation/precipitation; and through pervasive ferruginization of organic material, including plants and insects. Precipitation environment and rates and mechanisms of precipitation have a direct influence on goethite purity, crystallinity,
and suitability for (U-Th)/He dating (e.g., Vasconcelos et al., 2013; Monteiro et al., 2018b).

### 4.1. Colloform goethites

Colloform goethites (Fig. 2a) precipitate in empty spaces when iron species in solution interact with the exposed surfaces of host rocks, previously formed goethites, or other supergene phases (e.g., cryptomelane, malachite, etc.). Colloform
goethites display concentric growth bands showing clear bases and terminations; commonly, each band consists of goethite crystallites oriented in the growth direction. In some cases, a goethite band is followed by a different mineral band (e.g., Mn-oxides, hematite, Cu-carbonate) before resumption of goethite precipitation. Tens-of-centimeter-thick colloform goethite is common in weathered massive sulfide deposits, karst environments, and some weathered pegmatites. The size
and purity of colloform goethite suggest high concentrations of iron in solution. Colloform goethites provide ideal samples for geochronology due to their mineralogical purity and often the protracted history of iron precipitation they record.

### 4.2. Massive goethites





Massive goethites (Fig. 2b) lack of growth bands notable in colloform goethites. They constitute massive goethite formed by dissolution of primary phases and the ready local re-precipitation of iron from solution. Massive goethites may show complex textures associated with multiple nucleation sites, cross-cutting goethite generations often in veins, and the possible coexistence of the newly formed goethite with remnants of primary Fe-bearing minerals such as magnetite, hematite, or ilmenite. Massive goethites are commonly found in lateritized BIFs, basalts, continental sediments, and gossans.


### 4.3. Pore-filling goethites

Pore-filling goethites (Fig. 2c) in continental sedimentary deposits and their duricrusts cement together loose detrital grains. Generation after generation of iron-rich solutions penetrate pore spaces, often in waves, partially corroding detrital minerals

and precipitating bands of pore-filling goethite that may form Liesegang rings, concretions, and pisoliths. Pore-filling goethites also precipitate within and around root casts. In weathering environments, mixed ages may originate from the fact that several generations of pore-filling goethite coexist in a single ~400-500 μm grain.

### 4.4. Pisolitic goethite


Pisolitic goethite (Fig. 2d) consists of concentric layers of goethite surrounding a nucleus. They are found in many different geological environments and can be detrital or formed in situ. The nucleus of a pisolith may contain ferruginized sediments, rock fragments, another goethite pisolith, and pieces of broken pisoliths. Some ~1cm wide pisolitic goethites record growth histories ranging tens of millions of years (Lima, 2008; Hofmann et al., 2017), while others suggest rapid growth (Monteiro

et al., 2022).

### 4.5. Goethite replacing wood fragments

Preservation of plant and insect morphology in oxidizing environments is rare. However, rapid and pervasive goethite

replacement of wood fragments (Fig. 2e-f) and dead organisms may allow the complete preservation of plant cells and delicate soft tissues (e.g., McCurry et al., 2022). This phenomenon is common yet under-investigated in lateritized CIDs (e.g., Heim et al, 2006; Danisik et al., 2013) and other ferruginized fluvial or lacustrine systems. Porous (yellow) goethite replacing wood fragments or soft tissues may have poor He retentivity, imposing challenges for its use in (U-Th)/He





geochronology. On the other hand, goethite replacing tree logs (Fig. 2g) can be massive, very crystalline, and suitable for

(U-Th)/He geochronology.

### 4.6. Goethite biomineralization

Microorganisms play an important role in the biogeochemical cycling of iron in oxidizing environments (Monteiro et al., 2014; Levett et al., 2016). Biogenic goethites form when iron precipitates on the outer shell envelopes of microorganisms,

following by their eventual death and fossilization by the iron-rich solutions (Levett et al., 2016; Levett et al., 2020). The formation of duricrusts on lateritized BIFs is directly associated with the dissolution and reprecipitation of biogenic goethite cements (Monteiro et al., 2014; Levett et al., 2016, 2020). Biomineralization mimicking processes inferred in the formation of cangas have been reproduced in the laboratory (Levett et al., 2020).

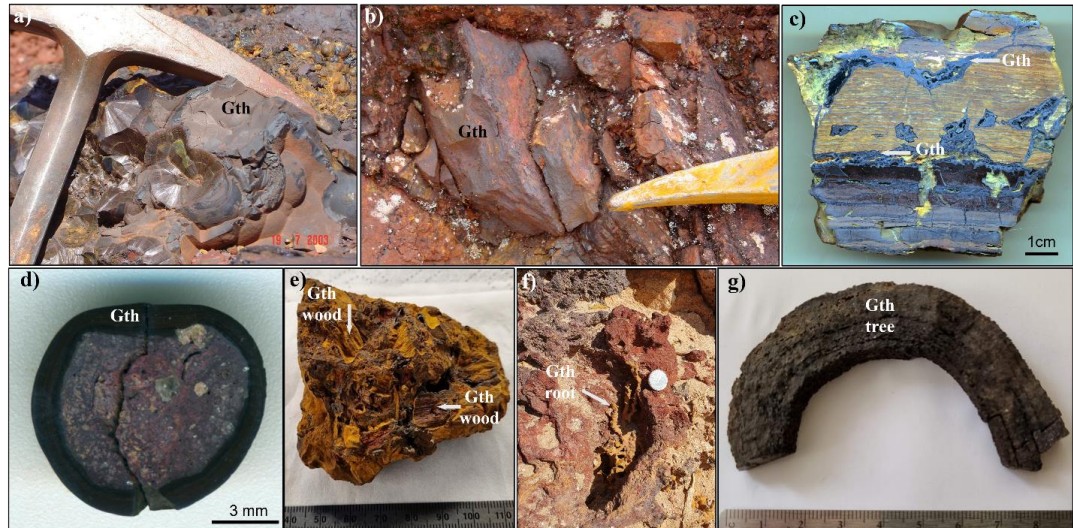


**Figure 2. Types of goethite dated by (U-Th)/He: a) colloform goethite; b) massive goethite; c) pore filling goethite or goethite cement; d) pisolitic goethite; e) ferruginized sediment hosting wood fragments replaced by goethite; f) yellow goethite forming root casts; and g) goethite replacing a tree log.**

**5.   U and Th concentrations in globally distributed goethites**





Uranium and thorium behave differently under earth's surface conditions. $U^{4+}$, $U(V)O_2^+$, and $U(VI)O_2^{2+}$ are commonly complexed in surface waters (Langmuir, 1978). The stability of different uranium complexes in weathering solutions depends on the oxidation potential and pH of the fluid, the amount of dissolved oxygen, the presence of sorptive agents, and the concentration of other species such as F⁻, Cl⁻, $CO_3^{2-}$, $SO_4^{2-}$, $PO_4^{3-}$, OH⁻, O²⁻, and organic complexing ligands
(Langmuir, 1978). Uranium is relatively mobile in oxidizing solutions, but $U^{4+}$ concentrations in weathering solutions are low because U tends to precipitate as insoluble uraninite and coffinite when solutions interact with organic matter (Langmuir, 1978). Massey et al. (2014) show that incorporation of U into the goethite structure may occur via reduction of U(VI) to U(V) in the presence of $Fe^{2+}$ in solution during ferrihydrite transformation to goethite. The prerequisite of an unstable precursor to goethite still needs confirmation. Reduction of $U(VI)O_2^{2+}$ to $UO_2$ and adsorption of $U(VI)O_2^{2+}$ on the
goethite surface are other retention pathways (Massey et al., 2014). In contrast, $Th^{4+}$ complexes are much less mobile and tend to remain very close to their mineral sources. Importantly, organic complexes of Th can be stable between pHs 4 and 8 (Boyle, 1982), increasing Th mobility in certain surficial environments (Monteiro et al., 2014, 2018).

Uranium and thorium concentrations in goethites can vary significantly within a weathering profile, and even within a single hand-sample. Variations in goethite U- and Th-contents are associated with changes in the composition of the source rocks,
local geochemical conditions, goethite precipitation mechanism, and the possible presence of microscopic inclusions (e.g., monazite) within the goethite grain selected for analysis. Commonly, goethites enriched in U are depleted in Th and vice-versa. However, goethites from some geological environments may be simultaneously enriched in U and Th (e.g., de Campos et al., 2023).

Figure 3 illustrates the concentrations of U and Th in goethites from various geological environments worldwide. The
protracted leaching process within long-lived duricrusts enriches surface goethites in Al and Th. This feature is particularly evident in samples from lateritized BIFs, as most U and Th measurements were carried out on goethites from cangas. The majority of the lateritized CID data was obtained for a single hand specimen from the Lynn Peak CID (Vasconcelos et al., 2013) and demonstrate a strong positive correlation between U and Th. Colloform goethites from weathered massive sulfide deposits often contain hundreds to thousands of ppm U but very little Th (Monteiro et al., 2018a). Similarly to cangas,
duricrusts blanketing massive sulfide deposits also contain goethites enriched in Th (Monteiro et al., 2018a). Most goethites contain less than 50 ppm U, except for goethites from massive sulfide deposits, lateritized alkaline-carbonatite complexes (ACC) and some continental sediments (Fig. 3). Most goethites from lateritized ACC plot within one of two distinct groups (U-rich, Th-poor or U-poor, Th-rich); a few goethites plot in-between (Fig. 3). Lateritized continental sediments contain the highest-Th goethites (up to ~765 ppm). The high Th contents of goethite cements and pisoliths reveal significant sources of
detrital Th minerals (e.g., monazite or thorite) in the sedimentary units and attest to the conservative behavior of Th with respect to U during recurrent in situ weathering of detrital phases. Goethites from quartzite karst consistently show higher concentrations of Th, while dolostone karsts show the opposite trend (high U). An extreme example of U- and Th-poor





goethite is the fossilized tree trunk goethite from the Springsure Coal deposit, Queensland, which consistently shows concentrations ≤ 0.1 ppm.


**Figure 3. Uranium and Th concentrations of goethites from nine distinct geological environments.**

## 6. The global distribution of goethite (U-Th)/He ages

Figure 4 illustrates histograms of the distribution of goethite (U-Th)/He ages from 100 to < 1 Ma for ten localities. Weathering environments in Brazil and Australia yield the most complete record of goethite crystallization ages. In Brazil, the frequency of goethite (U-Th)/He ages progressively increases after ~ 70 Ma, while in Australia an older group (~ 80-60 Ma) of goethites can be distinguished from a second group of goethites predominantly younger than ~ 30 Ma. French Guiana and Suriname show goethites that are predominantly younger than ~15 Ma and ~25 Ma, respectively, with many ages falling

between ~8 and ~ 2.5 Ma. The limited number of samples available for Canada (Artic circle) and China (Tibetan Plateau) show a narrow age range where most goethites are younger than ~ 5 Ma, while in Tunisia two distinct age groups are observed at ~9 Ma and ~ 1 Ma. Diagenetic pisolitic goethites from sediments of the Colorado Plateau in the USA yield a right-skewed distribution of ages from ~14 to ~ 2 Ma. In contrast, from 195 pisolitic goethite grains dated from a paleosol in Switzerland, only one age younger than ~10 Ma was obtained, with most ages falling between ~40 and 17 Ma. Goethites





from Morocco cluster at ~95-50 Ma and are among the oldest goethites in the world. Globally, young goethites are found
      in all investigated sites, except in paleosols from Switzerland and in karstic dolostones from Morocco. Goethite ages older
      than 100 Ma were obtained for two detrital goethite samples from colluvium deposits from the Quadrilátero Ferrífero, Brazil
      (Monteiro et al., 2020), and the Flinders Ranges, South Australia (Waltenberg, 2012).

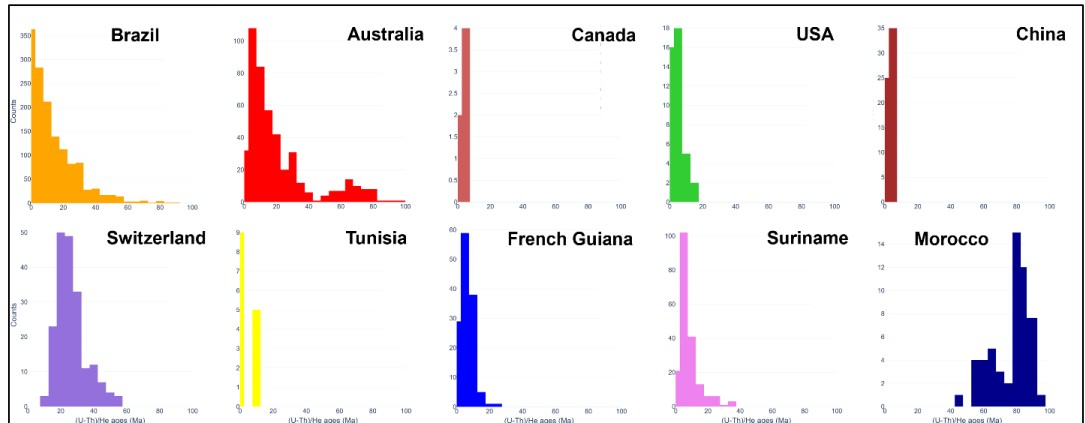


**Figure 4. Ten histograms representing the distribution of goethite (U-Th)/He ages across various regions globally, including Brazil, Australia, Canada, USA, China, Switzerland, Tunisia, French Guiana, Suriname, and Morocco. The histograms offer a detailed insight into the age distribution patterns within each region. Peaks or clusters in the histograms indicate predominant**

**age ranges, while the spread or dispersion of data points provides information on the variability of ages within each region. By combining geographical information with age distribution data, global trends and spatial variations in goethite (U-Th)/He ages show that geological processes and environmental factors influencing the formation and evolution of goethite-bearing weathering profiles differ across the planet.**

Figure 5 illustrates (U-Th)/He age (Ma) *vs* eU (ppm) for goethites in our compilation. Effective uranium is commonly used
      as a proxy for radiation damage in a mineral (Flowers et al., 2007). As previously documented for apatite, radiation damage
      may control He retentivity (Flowers et al., 2007). If He loss in nature is important, and if radiation damage makes goethites
      more retentive, a positive correlation between age and eU should be expected. No such correlation is observed for the global
      database of goethite ages, as illustrated in Figure 5, suggesting that radiation damage is not a factor controlling He retentivity

in goethite.

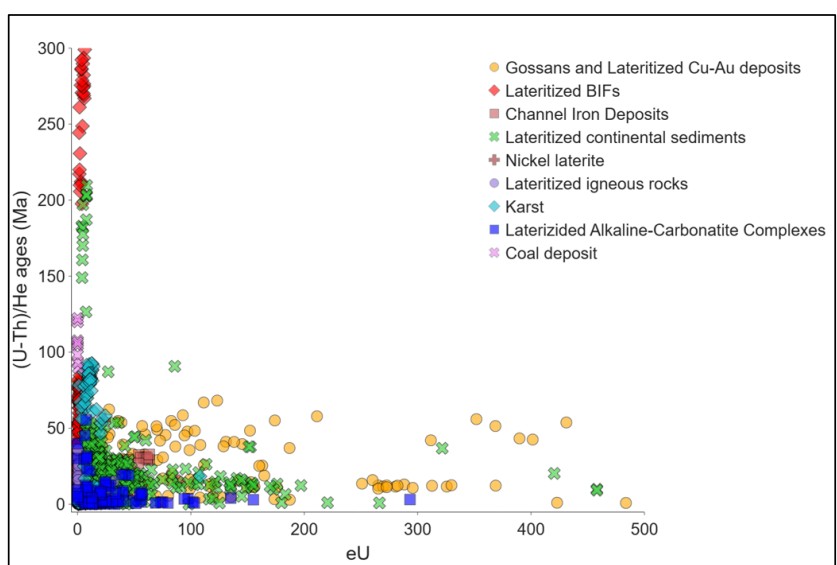

**Figure 5. Goethites from distinct weathering environments reveal varied eU concentrations and ages and lack of positive correlation between (U-Th)/He ages and eU. Noticeably, some of the oldest goethites contain significantly low eU concentrations, while relatively young goethites contain hundreds of ppm eU.**

## 7.  Summary

The global database of goethite (U-Th)/He ages and U and Th concentrations shows that a significant effort has been made by different research groups to select, characterize, and date goethite samples preserved in weathering profiles. Goethites from nine geological environments have been investigated. The oldest goethite ever dated comes from a colluvium deposit of lateritized BIF in Brazil (~ 284 Ma). However, most dated goethites are younger than ~66 Ma, with a significant increase in the frequency of ages younger than 23 Ma. Goethites from different environments show variations in U and Th contents.

U enrichment of 100s to 1000s of ppm are common in goethites from lateritized Cu-Au deposits, while similar enrichments in Th are observed for goethites from lateritized continental sediments. This dataset clearly shows that the chemical and isotopic compositions of dated goethites record information on changes in environmental conditions through time. The global distribution of goethite (U-Th)/He ages reveals that even though goethite is widely distributed over the surface of the Earth, an immense area of the globe known to contain goethite-bearing weathering profiles has not yet been investigated

using the (U-Th)/He method. The database also reveals that goethite geochronology applied to weathering studies is still in





its infancy, and that paleoenvironmental and paleoclimatic studies will benefit from the broader application of the (U-Th)/He method and other future methodological developments to goethite.

**Competing interests**


The authors declare that they have no conflict of interest.

**Author contribution**

Hevelyn Monteiro: investigation and writing – original draft preparation. Kenneth Farley: funding acquisition and writing – review & editing. Paulo Vasconcelos: writing – review & editing.

**Acknowledgements**

This research was supported by NSF grant EAR 1945974 to Kenneth A. Farley.



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
