# Peer review of "Short Communication: A database of the global distribution of (U-Th)/He ages and U, Th contents of goethites"

_Geochronology, 2024_

## Author Comment (AC1)

**Response to questions raised by anonymous Reviewer #1**

General comments

**Reviewer #1: "**This manuscript presents a summary of (U-Th)/He goethite ages stemming from 33 published studies and 12(?) unpublished studies around the world. The goal of the manuscript is to provide a database that can be used by referred to and added to by future studies.

However, while I see the value in providing such a database, particularly one that is accessible and can be added to in the future, I question whether the manuscript really contains enough information to form a publication without more information and some degree of interpretation of the results.

The manuscript would be a great resource to learn about all the environments goethite can form in and what dating it can reveal, e.g. deposition and weathering of sediments, or enrichment of ore deposits. However, it doesn't provide any insights into these processes using the data that has been compiled. Again, I can appreciate that for a short communication in a journal focused on geochronology, this might not seem important, but it felt to me that with this dataset more effort could be put into telling the reader what it shows."

As noted by the reviewer, the main objective of this manuscript is to compile and present a global database of goethite (U-Th)/He ages. The database will be available to anyone interested in contributing with published and unpublished results. Detailed interpretation of the results was deemed unnecessary because it was not our intention to write a review paper where we critically reviewed observations and conclusions of the published sources from which the databased was amassed. Interpretation and conclusions pertinent to each set of results can be retrieved from original publications. Nonetheless, a discussion section on the combined goethite (U-Th)/He results and their implication to earth's environmental evolution through time will be added to the revised manuscript.

**Reviewer #1: "**I also think a lot more information needs to be clarified from the start, particularly terminology that won't be familiar to all readers of Geochronology, such as terms like 'supergene'."

A list of terms and definitions will be added to the public github repository.

**Reviewer #1: "**The figures are quite poor and lack important information. They are also rather low resolution so it is quite difficult to discern key features within them, and often difficult to read labels and axes. They often lack labels and detail that would make them much more useful to the reader. I discuss each figure separately below."

The figures will be improved by the addition of proper labels, plot identifiers, etc.

**Reviewer #1: "**Overall, I think this database is a useful contribution to the community but I think it would benefit from a more thorough introduction and a bit more of an attempt to discuss the trends that the database reveals and what they might tell us about "changes in global environmental conditions through time" as mentioned in the Summary."

Suggestion accepted and to be implemented in the revised version of the manuscript.

**Reviewer #1: "**Specific comments

Clarify what is meant by "supergene" from the start. Surface weathering.... Deposits?"

The terms **supergene processes** and **supergene minerals** will be defined in a **List of terms and definitions** to be added to the revised manuscript.

**Reviewer #1: "**Line 23: Can you be more specific than "millions of years"?

We substituted "millions of years" by "tens to hundreds of millions of years".

**Reviewer #1: "**Line 26: Please give some examples of the "various settings"."

Added text: (e.g., weathered BIFs, ferruginized continental sediments, weathered basalts, karsts, etc.)

**Reviewer #1: "**Line 32: I think the concept of supergene enrichment of ore deposits needs to be introduced before it is raised in this sentence: "further show the influence of paleoclimate on weathering and supergene ore genesis"."

The term **supergene enrichment** will be defined in a **List of terms and definitions**.

**Reviewer #1: "**Line 33: "What are pisoliths?"

The term **pisolith** will be added to the **List of terms and definitions**.

**Reviewer #1: "**Line 39: "Without a more comprehensive introduction, it is unclear how the goethite (U-Th)/He database will elucidate changes in the frequency of precipitation, nor how that can be tied to environmental conditions."

We will expand the introductory text to clarify how goethite (U-Th)/He ages and chemistry can be used to study changes in past environmental conditions.

**Reviewer #1: "**Line 59: "Define "laterites" and "ferricretes". This could form part of a more comprehensive introduction before diving into the detail of the database. This would also help to explain why access to open-pit mining operations is helpful."

The terms **laterite** and **ferricrete** will be added to the **List of terms and definitions**.

**Reviewer #1: "**Figure 1: In the text, various country names are provided but they are not labelled on the map. Please label or highlight all countries discussed in the text, particularly those listed in Figure 4."

All countries highlighted in the text will be labeled and identified in Figure 1 using a similar color scheme as in Figure 4.

**Reviewer #1: "**Lines 72–73 (Fig. 1 caption): Add the age ranges for the Miocene, Pleistocene, and Pliocene groupings in parentheses so it is easier to compare them with the ages provided in the figure. Are the groupings in the figure related to Miocene, Pleistocene, and Pliocene? If not, explain why you have grouped them as you have. Be clearer that almost all of the ages are Cenozoic. Rather than

talking about 65 Ma, be clear that this is the Cenozoic/Mesozoic boundary, so very few are Mesozoic (big climate changing event at this boundary?).”

Added text: Miocene (5.3 – 23 Ma), Pliocene (2.6 – 5.3 Ma), Pleistocene (2.6 – 0.01 Ma).

Substitute *“(U-Th)/He ages older than 65 Ma [...].”* with *“Goethites of Mesozoic (66 – 252 Ma) and Paleozoic (252 – 539 Ma) ages only occur in Brazil [...], Australia [...], and Morocco.”*

**Reviewer #1: “**Line 89: Briefly tell us what banded iron formations are.”

The term **banded iron-formations (BIF)** will be added to the **List of terms and definitions**.

**Reviewer #1: “**Line 93: Give formulae of hematite and magnetite. Give examples of iron-bearing carbonates. Also might be worth mentioning that hematite and magnetite can and have also been dated using (U-Th)/He method but that you are just focusing on goethite here?”

Added text: “Hematite ($Fe_2O_3$) and magnetite ($FeO.Fe_2O_3$) [...] iron-bearing carbonates (siderite ($FeCO_3$), ankerite ($(Ca,Fe)CO_3$, etc.) [...].”

**Reviewer #1:** Line: 98: Explain the concept of “lateritized”. A brief introduction to laterization would be a useful part of an expanded introduction.”

The term *lateritization processes* will be added to the **List of terms and definitions**.

**Reviewer #1: “**Line 100–101: What do you mean by “absolutely” and “relatively” iron enriched? Can you clarify?”

The concepts of absolute and relative *supergene enrichment* will be described in a **List of terms and definitions**.

**Reviewer #1: “**Line 103: When you say the primary hematite is inherited from the bedrock, do you mean it is a primary within it? If so, maybe just say “primary hematite within the bedrock” rather than “inherited from it”?”

The sentence will be modified to read “In contrast, primary hematite dominates in the lower horizons, [...]”.

**Reviewer #1: “**Lines 103–104: This is confusingly worded. I would rewrite this section along the lines of: “Goethite dominates the mineralogy of cangas (Monteiro et al., 2014), whereas older, primary hematite within the bedrock dominates in the lower horizons (Monteiro et al., 2018a), and goethite is restricted to veins or local replacement of minor carbonates or silicates (e.g. siderite or grunerite).”

We thank the reviewer for the suggestion.

**Reviewer #1: “**Lines 127–128: What does it mean if iron cementation “intensifies”? Do you mean the abundance of goethite increases? And how does that relate to no goethites younger than ~5 Ma? Presumably the lack of goethites after ~5 Ma is counter to this intensification? If so, maybe replace “notably” with “although” or “however”.”

Intensification of iron cementation means that the warm and wetter conditions in the Miocene and the availability of large amounts of organic acids related to the decomposition of organic material in

the channels favored the dissolution of iron-oxides in BIFs and the reprecipitation of iron as goethite cements. In their original papers, Heim et al. (2006) and Vasconcelos et al. (2013) interpreted the decline in ages younger than 5 Ma as resulting from the progressive aridification of the Australian continent from the Pliocene onward.

**Reviewer #1: "**Lines 131–132: What are massive sulfide deposits and iron-oxide copper gold deposits? (presumably IOCGs are a type of massive sulfide deposit?) These could be briefly introduced in a revised Introduction, as suggested previously. Presumably it's the Fe sulfides (e.g. pyrite) producing the Fe for the goethite? But this isn't made clear."

The terms *massive sulfide deposits* and *iron-oxide copper gold (IOCG) deposit* will be added to the **List of terms and definitions**.

**Reviewer #1: "**Line 141: Weathering of which sulfides will produce Fe2+ and Cu2+? Does the sulfide being weathered have any bearing on what Fe-oxide is produced (goethite vs hematite?)."

$Fe^{2+}$ weathers from magnetite ($Fe_3O_4$) and pyrite ($FeS_2$), and $Cu^{2+}$ and $Fe^{2+}$ from chalcopyrite ($CuFeS_2$). The type of supergene iron oxyhydroxide is determined by conditions such as water activity, pH, and surface temperature. Weathering of sulfides will decrease the solution's pH, which favor the precipitation of goethite.

**Reviewer #1: "**Line 188: You mention here "geochronology of goethites and hematites" but this is the first time mention has been made of dating hematite. The potential to (U-Th)/He hematite, and how this compares with goethite, should be raised earlier."

We thank the author for the suggestion. We will address geochronology of both supergene goethite and hematite in the revised Introduction of the manuscript.

**Reviewer #1: "**Line 201: It would be good to see this area on a map. Or at least say how far it is from the mouth of the Amazon River to Rio de Janeiro?"

We thank the author for the suggestion. We will add these geographical references to the revised version of the figure.

**Reviewer #1: "**Lines 230–231: What is the significance of the different coloration? They give different ages, so presumably their colors are meaningful? Do they get darker each time they are re-weathered? Is that why black ones are the youngest group…?"

The coloration of the different types of goethite is related to their chemical composition and crystallinity. Well-packed large crystallite goethites will appear darker than porous, small crystallite goethite masses. We will add this information to the revised version of the manuscript.

**Reviewer #1: "**Line 265: Replace "of" with "the" so the sentence reads "Massive goethites lack the growth bands notable in colloform goethites".

We thank the author for the suggestion.

**Reviewer #1: "**Figure 2 doesn't do justice to the range of goethite textures discussed in the text. To get a true understanding of how goethite presents itself, textural relationships with other phases, and implications for dating (e.g. grain size, mixing with other U-Th-bearing minerals), there should be thin

section and, ideally, SEM images showing some examples. And where are these samples from? Are they samples that form part of the database?"

We will make a new figure with photomicrographs and SEM images of goethite grains showing different textures and paragenetic relationships with other minerals.

We will clarify that all samples in Figure 2 were dated by (U-Th)/He and their ages are included in the database.

**Reviewer #1: "**Line 277: Can you show an illustration of a single grain with multiple generations of goethite? Is anything visible in such a grain to tell? Zonation in SEM? If there is no way to tell prior to dating, this is also useful information."

We will add an image of a goethite grain showing multiple generations in a new figure to be added to the manuscript.

**Reviewer #1: "**Line 284: How rapid is rapid? Can you be more specific? And is this based on textural relationships/thin rims or (U-Th)/He dating?"

We will define what constitutes rapid goethite precipitation in the revised version of the manuscript.

**Reviewer #1: "**Line 286 (section 4.5 title): Since both are discussed, should this title be "Goethite replacing wood fragments and soft tissue organisms"?"

We thank the reviewer for the suggestion.

**Reviewer #1: "**Line 316: Give formulae of uraninite and coffinite."c

We will provide formulas for all minerals in the revised version of the manuscript.

**Reviewer #1: "**Line 325–326: The possible presence of microscopic U-Th-bearing minerals, such as monazite, should be addressed previously in the manuscript. How big an issue can it be? How can it be avoided? (again, an improved Fig. 2 showing SEM images would help to explore this issue)."

In the revised version of the manuscript, we will add a paragraph to address possible complications related to goethite crystal sizes, porosity, multiple generations, chemical composition, and possible contamination by other minerals, such as monazite, zircon, hematite, quartz, etc.

**Reviewer #1: "**Line 327: Which geological environments are likely to have simultaneously enriched U and Th contents?"

We will add a section in the revised version of the manuscript that describes the geological environments likely to drive simultaneous enrichment of U and Th in goethites.

**Reviewer #1: "**Lines 339–341: "The high Th contents of goethite cements and pisoliths reveal significant sources of detrital Th minerals (e.g., monazite or thorite) in the sedimentary units...." Are you referring to Th that has been remobilized from monazite and thorite and incorporated into the goethite as it grew or inclusions of monazite and thorite grains incorporated within the goethite? This is unclear but has potential implications for (U-Th)/He dating if such inclusions are not accounted for."

We are refereeing to Th leached from monazite and thorite and later incorporated into goethite. We will clarify this issue in the revised version of the manuscript.

**Reviewer #1:** Figure 3: It is very difficult to discern the axes on the plots but I am assuming they are Th (ppm) vs U (ppm). I suggested labelling each panel (a) – (i) so they can be referred to more clearly in the text."

We thank the reviewer for the suggestion.

**Reviewer #1:** "Lines 357–358: The USA histogram in Figure 4 doesn't look "right-skewed" to me, although I'm not certain what "right-skewed" means in this context."

Skewness describes the symmetry of a distribution. A distribution is right-skewed if it has a tail on the right side of the distribution curve.

**Reviewer #1:** "Lines 359–360: "Goethites from Morocco cluster at ~95-50 Ma..." but there appears to be a bimodal population, with a dip around 75 Ma. How do you account for that? Or, if you can't account for it, then at least make note of it."

Agreed. We will make the necessary changes in the text.

**Reviewer #1:** "Line 360: When you say "Globally, young goethites", how do you define "young"?"

In the present manuscript, we define young as ages younger than ~2.6 Ma.

**Reviewer #1:** "Lines 386–387: "Some of the oldest goethites contain significantly low eU concentrations while relatively young goethites contain hundreds of ppm eU". Any thoughts on why this might be?"

The variation in eU concentrations rise mainly from the uranium and thorium concentrations of the different lithologies and the environmental conditions where goethite precipitated. For example, in the Igarapé Bahia site, most goethites precipitated at depth are enriched in U (100 – 1000's ppm) and depleted in Th (mostly < 1 ppm) and yielded (U-Th)/He ages varying from ~ 60 to ~1 Ma. Goethites precipitated near the surface show high Th (~3 - ~110 ppm) and low U (~3 - ~20 ppm) concentrations and ages varying from ~42 to ~12 Ma. In general, as illustrated in our compiled database, there is not trend between age and effective U concentrations.

**Reviewer #1:** "Line 402: "Future methodological developments" is rather vague. Such as? Along what lines?"

Examples of future methodological developments are in situ (U-Th)/He dating and in situ analysis of stable oxygen isotopes using ion microprobes. We will outline those developments in the revised manuscript.

**Reviewer #1:** "Technical corrections

Line 38: Remove the word "aim" and just say that you "assess the influence...".""

Suggested modification will be implemented.

**Reviewer #1: "**Try not to switch between present and past tense throughout the manuscript, e.g. Lines 47–48 "For cases where the mass of the analyzed grain was not provided and parent element amounts are reported, it was impossible to calculate concentrations in ppm""

We will address possibly inconsistent use of verb tense in the revised manuscript.

**Reviewer #1: "**Line 50: Rather than "bedrock", perhaps better to say "lithology"?"

Both the terms bedrock and lithology are used in the geological literature.

**Reviewer #1: "**Line: 57: Change "map distribution" to "spatial distribution"."

We thank the reviewer for the suggestion.

**Reviewer #1: "**Line 61: Replace "similar geographic positions" to "similar latitudes"."

We thank the reviewer for the suggestion.

**Reviewer #1: "**Line 70: In the Figure 1 caption, I would perhaps qualify the first sentence to say "...distribution of dated goethites included in this study" in case you missed any."

We thank the reviewer for the suggestion.

**Reviewer #1: "**Line 74: Label "Amazon and Quadrilatero Ferrifero" and "Hamersley Province and Flinders Ranges" on the map in Figure 1."

We thank the reviewer for the suggestion.

**Reviewer #1: "**Line 80: Define "duricrust"."

The term *duricrust* will be added to the **List of terms and definitions**.

**Reviewer #1: "**Line 99: Refer here to Fig. 1, where you need to have labelled these specific locations."

We thank the reviewer for the suggestion.

**Reviewer #1: "**Line 101: replace "transition" with "transitions"."

We thank the reviewer for the correction.

**Reviewer #1: "**Line 101: Define "saprock" – either here or in the introduction.

The term *saprock* will be added to the **List of terms and definitions**.

**Reviewer #1: "**Line 106: What is "hypogene"? – another definition for the introduction."

The term *hypogene* will be added to the **List of terms and definitions**.

**Reviewer #1: "**Lines 102–114: Monteiro (2014) and Monteiro (2018a) are cited 7 times within these 12 lines. Try to consolidate them so it isn't so repetitive."

We thank the reviewer for the suggestion.

**Reviewer #1: "**Line 130 (and 139): Define "Gossan". Since this is typically applied to ore deposits, it's important to explain it. Add it to the broader introduction."

The term *gossan* will be added to the **List of terms and definitions**.

**Reviewer #1: "**Line 138: Define "mottled zone"."

The term *mottled zone* will be added to the **List of terms and definitions**.

**Reviewer #1: "**Line 145: Replace "too fine grained and not suitable...." with "too fine grained to be suitable...""

We thank the reviewer for the suggestion.

**Reviewer #1: "**Line 150: "Goethites precipitated at depth" – what kind of depth?"

We will modify the sentence to say "Goethites precipitated below ~50 m depths".

**Reviewer #1: "**Line 161: Define "limonite"."

The term *limonite* will be added to the **List of terms and definitions**.

**Reviewer #1: "**Line 189: "targeting samples on Mars..." rather than "...at Mars"."

We thank the reviewer for the suggestion.

**Reviewer #1: "**Line 220: Typo – Should be "Morocco" rather than "Morroco"."

We thank the reviewer for the correction.

**Reviewer #1: "**Line 294: Remove "very" from "very crystalline"."

We thank the reviewer for the suggestion.

**Reviewer #1: "**Line 300: Replace "following" with "followed"."

We thank the reviewer for the suggestion.

**Reviewer #1: "**Line 332: Replace "data was obtained" with "data were obtained...". And remind the reader that Lynn Peak is in Australia."

We thank the reviewer for the correction.

**Reviewer #1: "**Line 334: If you give a value for U then you should also give a value for Th (rather than just "very little")."

We thank the reviewer for the suggestion.

**Reviewer #1: "**Line 338: "A few goethites plot in-between (Fig. 3)" – I suggest labeling the panels (a) to (i) so they can each be referred to more clearly at various points within the text. And in this instance, point specifically to where the "in-between" data lies so it is clear to the reader."

We thank the reviewer for the suggestion.

**Reviewer #1: "**Lines 343–344: Is this fossilized tree trunk goethite the same as shown in Fig. 2g? Or at least from the same locality? If so, refer to the figure."

Yes, it is the same sample. We will clarify this point in the revised version of the manuscript.

**Reviewer #1: "**Line 355: Correct "Artic" to "Arctic"."

We thank the reviewer for the correction.

**Reviewer #1: "**Lines 376–377: Try to rephrase these two sentences so that you're not citing Flowers et al. (2007) twice in quick succession."

We thank the reviewer for the suggestion.

The Excel database:

In column A (Authors), provide year as well. Can you add a link to each reference as well for easy access?

Yes, we will add the links for each reference.

For Calculated Age, what level of uncertainty is being reported? 2 sigma?

We report results at 1 sigma. We will clarify this issue in the revised version of the manuscript.

---

## Author Comment (AC2)

**Response to questions raised by anonymous Reviewer #2**

**Reviewer #2:** "This manuscript provides a database of (U-Th)/He ages (almost 3000 ages) and U-Th contents of goethite samples from different countries and weathering environments in the world. Authors attempt to assess the influence of environmental conditions on changes in the frequency of precipitation and preservation of goethite in weathering profiles. I agree with the authors that their data set may contribute to understanding goethite (U-Th)/He dating of supergene weathering processes. I think this work is suitable for Geochronology, and they have interesting data set to contribute. However, in the paper's current form, I think with a modest and careful effort to revise the manuscript, this will make a nice addition to this journal. My only concerns are: (1) Authors provide a large amount of information on ages and U-Th contents of goethite, but discussion and interpretation are limited; (2) the figures need to be improved; and (3) implication on paleoenvironmental and paleoclimatic studies."

We thank the reviewer for encouraging the publication of our manuscript. To fulfill the reviewer's requests, we will:

1. Create a new discussion section with interpretation of the (U-Th)/He results.
2. Improve figure quality by adding labels, increasing font size, coloring of countries in the map (Figure 1) for which (U-Th)/He data are available. In addition, a new figure will be created with photomicrographs and/or SEM images of different types of goethites.

**Reviewer #2:** "1. "Introduction" - This section authors proposed that goethite dating can provide information on global environmental conditions, but they do not clearly explain the links between goethite formation and environmental conditions. I also suggest authors may add some explanation on why goethite is important and profit to (U-Th)/He dating; why other iron-oxides (lepidocrocite, hematite, magnetite, limonite) are not widely used to (U-Th)/He dating? Is it related to well preservation, crystallinity, high U-Th contents, or specific paleoclimatic condition?"

Goethite precipitation depends on solution chemistry, $H_2O$ availability, pH, and temperature. Temperature and pH have a greater effect on precipitation rates. Ph also plays a role in the incorporation of other cations (e.g., Al) in the goethite structure. The literature about the conditions under which goethite precipitates is vast, and we will introduce some of the most important ideas and the implications to the use of goethite as an environmental marker.

We will also expand our introduction section to explain why goethite until now dominates in studies of surface processes that applies the (U-Th)/He method.

**Reviewer #2:** "2. "Geological Environments" and "Types of goethite"-Authors spend the great length to describe the geological environment, occurrence, geochemical characters, and Eh-Ph of the goethite. I think those two sections should be combined and reduced, if authors add a summary table that list the location, elevation, profile, depth, paleoclimatic conditions, goethite occurrence (colloform, massive, infilling, pisolitic....), crystallinity, mineralogical associations (goethite+hematite+clay, goethite+magnetite+ilmenite, goethite+Mn-oxides, goethite+gibbsite+gold, goethite+malachite+azurite+cuprite+native copper+chalcocite, ....), geochemical composition (Al, Cu, Ni, P, REE, Co, U, Th, U/Th...), and (U-Th)/He ages of different

geological environments, then it is easy to compare the goethite in different geological environments."

Thank you for the suggestion. We will summarize the relevant information in a table.

**Reviewer #2:** "3. "U and Th concentrations in globally distributed goethites"- The U-Th concentrations in goethite are very interesting, I hope author may give more interpretation of these results. First, authors only use the reported data in ppm to plot, thus some information may loss. In this case, I suggest authors may plot with element molar ratios, such as U/Sm, Th/Sm, and U/Th...Because we find the U and Th concentrations are variable within a weathering profile and even within a single hand-sample, but U and Th commonly show a strong positive correlation, this indicates they have similar U/Th ratios in certain surficial environments, thus it possibly reflects the source of weathering rocks or others. Second, I suggest U vs Th concentrations plates of distinct geological environments may plot in a single plate, it is easy to compare the U-Th distributions at different condition. I also suggest authors may add U/Th ratios vs deep or Sm/Th ratios, and then compare and discuss the geochemical characters of goethite in different geological environments."

We thank the author for the suggestions. We will incorporate more information on U and Th contents in the new version of the manuscript.

**Reviewer #2:** "4. "The global distribution of goethite (U-Th)/He ages"-The (U-Th)/He age distribution and age vs eU plots of goethite samples from distinct environments are significant. First, the age distribution plots (Figure 4) show the ages from the most regions (Canada, USA, China, Tunisia, French Guiana, Suriname) are almost younger than 20 Ma, some regions (Brazil, Australia, Switzerland) show goethite formed since >60Ma, but goethite from Morocco formed at 40-100 Ma. I expect authors do not just show the age distribution, but add some explanation on why these regions have distinct age distribution patterns. Second, although eU vs (U-Th)/He age plots do not show positive correlation, goethite with old ages (>~80Ma) have very low eU, especially samples from lateritized continental sediments show low eU in the old age sample but high eU in the young age sample. Is it possible that high eU goethite may damage the mineral's structure, and thus the He loss of high eU samples cause the young age or U loss of sample cause the goethite have low U and high ages?"

Again, we thank the author for the suggestions. We will discuss the global distribution of (U-Th)/He goethite ages in more detail in the revised version of the manuscript.

The lack of positive correlation between eU (U + 0.235 * Th) concentration and ages suggest that radiation damage is not a significant factor controlling He retentivity in goethites. Crystallite size distribution appears to be the main factor controlling He release from goethite. Now, the amount of U and Th in a sample will depend on the availability of these cations in solution. For example, multiple cycles of goethite dissolution in oxidizing environments will favor U mobilization towards lower horizons in the weathering profile. Consequently, the young goethites precipitated at depth will become enriched in U.

**Reviewer #2:** "5. "Summary"-I expect to what kind of goethite samples are suitable for (U-Th)/He. Also, authors may add some approaches for goethite (U-Th)/He age interpretation in future works; such as, more geochemical works (such as U, Th, Al, P, Si, Ti, Ni, Cu, REE...) on goethite is benefit to

understand weathering processes, the researches on the crystallinity of goethite are used to evaluate the He retentive, and stable isotopes (C, N, H, O, Fe, Cu, Zn...) may reflect the paleoclimatic conditions during the goethite formation."

Once again, we thank the author for the suggestions. We will discuss the issues suggested above in the revised version of the manuscript.

---

## Author Response (AR1)

**Associate editor decision: Publish subject to revisions (further review by editor and referees)**

Both reviewers think that your short communication on "A database of the global distribution of (U-Th)/He ages and U,Th contents of goethites" is suitable for publication in Geochronology pending revisions. Having read your manuscript myself, I agree with this assessment, although I have a slightly different opinion about the nature of the revisions. The main problem with your manuscript is that it is too long for a short communication and too short for a review article. Therefore, I would like to give you the choice between two types of revision:

1. You can follow the reviewers' recommendations and turn your manuscript into a bona fide review article by expanding it with further background information and definitions. Such a revised manuscript would need to be resubmitted with a different title and re-reviewed.

2. Alternatively, you can shorten the manuscript and stay true to its current title. Short communications are meant to be simple announcements of methodological advances that may be useful to other geochronologists. Thus, you can focus on the database and remove the detailed methodological discussion.

Regardless of which option you choose, I would recommend that you make your database more user-friendly by linking it to an interactive web-platform. In its current form, your database consists of a number of csv files that have been posted on GitHub. The landing page for your GitHub page provides instructions on how to clone the repository, but does not encourage users to issue pull requests, nor provide a mechanism to ensure that the database will be maintained in the future.

One easy way to make your database available to the wider community is to upload your data to existing databases such as the AusGeochem web-portal, which will change its name to EarthBank in the near future, reflecting its global scope. This database includes formats for U-Th-He data. Once you have uploaded your data to this web-portal, you will be able to use the built-in data visualisation tools to enhance the figures in your manuscript.
* * *
**Reply to Associate Editor**

Dear Peter Vermeesch,

We would like to thank you for your suggestions on how to proceed with this manuscript. To publish this work as a Short Communication article, we replaced two sections of the manuscript ([section 3. Geological Environments] and [section 4. Types of goethite]) with a much shorter section ([section 3. Weathering Environments and Types of goethite]) that briefly summarizes key information on weathering environments studied with the formation of goethite suitable for (U-Th)/He geochronology. As requested by the reviewers, we added a Discussion section where we highlight how goethite weathering geochronology helps to unveil the long-term history of landscape evolution of continental landmasses.

We also decided to maintain the database available as a repository in GitHub. Detailed instructions on how to use the database and how to contribute with new data have been described in the Contributing section of the README file.

**Contributing**

Contributions from the community are welcome! Whether it is a typo error fix, data additions, or improvements to documentation, your help is appreciated. Please follow the guidelines below to contribute to this project.

1. Fork the repository (Fork a repository - GitHub Docs).
2. Clone the forked repository (Cloning a repository - GitHub Docs).
3. Create a new branch for your new entry.
4. Make your changes.
5. Commit your changes.
6. Push your changes to your fork.
7. Create a new Pull Request.

**Contact**

For any questions or suggestions, please contact Hevelyn Monteiro at hevelynbr@gmail.com.

Finally, we contemplate making the database available through Caltech webpage in the near future.

Sincerely,

Hevelyn Monteiro

---

## Author Response (AR2)

**Response to Reviewer #1 comments:**

This revised version of this manuscript has been amended to make it suitable for a short communication rather than a review paper, so I note that many of my original comments about expanding the discussion are unwarranted. I think this version is improved and I appreciate the efforts of the authors to address many of my previous comments. However, I am confused that quite a few of the authors' responses do not align with the revised manuscript – i.e. they state that they will make a certain change but the change has not happened. There are other responses that just say "We thank the reviewer for the suggestion" without stating what (if any) change was made.

Correct, the manuscript was modified and condensed for publication as a *Short Communication* paper and a *Discussion* section was added to the revised version of the manuscript.

In our previous *Response to the Reviewer Comments*, we replied with "We thank the reviewer for the suggestion" to comments simply requesting rewording of sentences, a change of a subtitle, or correction of spelling mistakes. Despite not specifying exactly the change to be applied, in most cases, the reviewer requests were implemented. Suggestions regarding the redesign of figures were carefully considered and implemented in the new version of the manuscript.

Below I summarize the main issues I have identified, including ones that have supposedly been addressed but not according to the revised manuscript I am looking at. However, I ask the authors to carefully double check their responses to make sure they have done what they said they would.

1. Many of the terms that have been put into the glossary on github would be better placed in the main text, at least some of the shorter ones. Also, not all the terms I queried originally have been included, despite the response stating that they "will be added to a List of terms and definitions" – e.g. banded iron formations, "absolute" and "relative" supergene enrichment, iron ore copper gold deposits, massive sulphide deposits, saprock, mottled zone. Please carefully check again to ensure all definitions are given within the text or in the glossary.

The following definitions were added to the Glossary: *absolute and relative enrichment* (as part of the definition of Supergene processes), Banded iron-formation, Iron-oxide Copper Gold (IOCG) deposit, massive sulfide deposit, mottled zone, saprock, saprolite.

2. In Figure 1, the authors said they would put the age ranges in the figure caption so the geological time periods could more easily be compared with the numbers on the figure. Yet this has not happened.

Figure 1 no longer shows information on the age range. Following requests for simplification of Figure 1, we decided to show clusters of points (ages) at the different regions.

3. Figures 1 & 5 – It seems odd to group the ages by country. Since geology has no geographic boundary, what is the significance of showing the age distribution of, for example, Morocco relative to Tunisia? In Figure 1, I don't understand the legend. It seems to suggest that all

Australian samples (red) have ages of 251.9-299 Ma (?) Or are the colors of the circles separate to the colors on the map? If so, come up with a different scheme to avoid confusion. In Figure 5, make sure axes on the subplots are labeled.

We deem it important to show the number and range of ages per country to highlight how little is known about the age of weathering profiles within each region and to instigate the interest of other researchers in the application of the method in regions with little or any data at all. We understand that geology does not regard country boundaries, but geological investigation is bound by geopolitical issues (research capabilities, funding, access to sites, etc.). Throughout the manuscript, we present and discuss changes in goethite ages and U- and Th-contents regarding weathering environments.

We dramatically simplified Figure 1 and made the necessary modifications to Figure 5.

4. The use of colors is confusing because they don't hold for the other figures – e.g. in Figures 3 and 6, it looks like red should be Australia but presumably it's data from more than one country? If so, this needs to be made clear.

The new figures are in black and white to avoid confusion.

5. I agree with the editor's suggestion that this dataset would be of most use to the community if it was uploaded to an existing database rather than within a spreadsheet on github.

Access to the GitHub page is straight forward. Anyone can copy, add, edit, and make comments about the database using the GitHub link. I also encourage anyone interested to contribute to the database to contact me at hevelynbr@gmail.com for questions and discussion of future collaboration.

I hope these comments are helpful in finalizing the manuscript for publication.

Thank you for your valuable suggestions and the time and effort you invested in improving the manuscript.